# One-Stage Tricompartmental Hypoallergenic UKA for Tricompartmental Osteoarthritis: A Case Report

**DOI:** 10.3390/healthcare11222999

**Published:** 2023-11-20

**Authors:** Andrea Parente, Marta Medetti, Giuseppe Basile, Franco Parente

**Affiliations:** 1Hip and Knee Replacement Department, IRCCS Orthopedic Institute Galeazzi, 20161 Milan, Italy; andreaparente320@gmail.com (A.P.);; 2Legal Medicine Unit, IRCCS Orthopedic Institute Galeazzi, 20161 Milan, Italy; basiletraumaforense@gmail.com

**Keywords:** osteoarthritis, knee replacement, UKA

## Abstract

Osteoarthritis (OA) is a degenerative and progressive joint disease. When all three compartments are involved, end-stage OA is treated with a total knee arthroplasty (TKA). Unicompartmental knee arthroplasty (UKA) is a primary treatment for isolated osteoarthritis. UKA has a quicker recovery time than TKA, as well as less morbidity and more tissue sparing. At the time of surgery, 17% of patients have a tricompartmental disease and most patients with a Kellegren–Lawrence grade >3 have an intact anterior cruciate ligament (ACL). Conventional TKA sacrifices the ACL. Patients with concurrent medial and lateral osteoarthritis and a functional ACL may receive a primary bi-unicondylar arthroplasty. Combined partial knee arthroplasty (CPKA) is an established practice either in bicompartmental femoro-tibial OA or in OA progression after UKA, with the addition of another UKA. A conversion of a lateral UKA to a tricompartmental joint replacement has been reported in the literature. In our case report, we describe a one-stage hypoallergenic tricompartmental UKA, with improved clinical score and no sign of early failure at the last follow-up.

## 1. Introduction

Osteoarthritis (OA) is a degenerative and progressive joint disease, which affects 250 million people worldwide [1]. Symptomatic OA of the knee affects up to 10% of men and 13% of women aged above 60 years [2].

Knee OA can be defined clinically and radiographically. Kellgren–Lawrence (KL) radiographic grading system is the most common radiographic method used. This scoring system classifies OA into five grades from 0 to 4, with grade 0 defined by no radiological findings of OA to grade 4 with deformity and loss of joint space [2]. Symptomatic knees sometimes have no radiographic sign of OA, while some Kellgren–Lawrence grade 3 can be asymptomatic. Diagnosis of knee OA must be conducted using a variety of clinical and radiological methods [3].

Standardized and recommended available treatments have been developed by scientific societies [3,4]. Intra-articular (IA) injections have a more direct effect with fewer systemic complications. Multiple substances have been delivered. IA injections of corticosteroid provide a reduction of the pain in the short term in OA. Glucocorticoid injections efficacy is higher than other type of drugs [5], reducing episodes of acute pain and increasing joint mobility [6]. On the other hand, Handler and Wright outlined radiographically the destruction of knee joints and cartilage after corticosteroid injections [7].

In the synovial fluid of an osteoarthritic knee joint, the molecular weight of hyaluronic acid (HA) falls by 33–50% [8]. HA injections improve viscosity, shock absorption, and joint elasticity through viscosupplementation [9]. Injection of platelet-rich plasma (PRP) turned out as a good treatment option. Injection of platelet-rich plasma seems to be more effective in young patients in the early stage of OA, with a similar efficacy as HA [10].

Conservative treatment can be very useful in mild to moderate knee osteoarthritis. Pain and altered physical function can be treated effectively with physiotherapy through therapeutic exercise, based on quadriceps strengthening and aerobic exercise, improving range of motion and proprioception. Dry needling (DN) used in combination with physiotherapy can produce significant improvement in pain, range of motion, and function in patients with persistent pain after TKA. To improve pain and function in painful knee OA electrical DN in a manual therapy can be added to the exercise program [11].

Symptomatic and advanced OA, not responding to conservative treatment affecting daily life, has to be managed with surgical treatment [12]. The correlation between OA radiological evidence and patients’ symptoms determines the time point of surgery. Arthroscopic lavage and debridement had been proposed [13], in order to relieve symptoms by eliminating inflammatory cytokines and debris. Anyway, arthroscopy’s role in knee OA is debatable [14]. There is a lack of demonstrating real benefit [15]. Only patients with evident meniscus lesions or cartilage flaps may have an improvement from surgery [15]. On the other hand, cartilage repair techniques were developed. However, cartilage repair can be carried out only in case of focal defects, because damaged cartilage has a limited capacity for healing [16].

Knee OA can involve in combination or individually all three compartments [17]. Traditionally, when all three compartments are involved, end-stage knee OA is treated with a total knee arthroplasty (TKA) [18].

Unicompartmental knee arthroplasty (UKA) is a primary treatment for isolated osteoarthritis (OA) [19], by the use of medial or lateral unicondylar or patellofemoral implants.

UKA has a quicker recovery time than TKA, less morbidity and more tissue sparing [19]. At the time of surgery, 17% of patients have a tricompartmental disease and most patients with a Kellegren–Lawrence grade >3 have an intact anterior cruciate ligament (ACL) [20].

Conventional TKA sacrifices the ACL with a bit lower function and increased laxity [19], but patients with concurrent medial and lateral osteoarthritis and a functional ACL may receive a primary bi-unicondylar arthroplasty (Bi-UKA) [21]. Combined partial knee arthroplasty (CPKA) is an established practice [21,22]. On the other hand, one of the most common reasons for the revision of UKA is the OA progression: a revision with the addition of another UKA is described in the literature [21,22,23]. In a past study, the conversion of a lateral UKA to a tricompartmental joint replacement with retention of both cruciates had been described [19].

In our case report we describe a one-stage hypoallergenic tricompartmental UKA following CARE (CAse REport) criteria [24].

## 2. Case Description

A 70-year-old woman came to our clinic with knee pain, resistant to non-steroidal anti-inflammatory drugs. She was an active woman, swimming and bicycling twice a week. In the past years, she underwent previously at multiple injections of high molecular weight hyaluronic acid and after platelet-rich plasma (PRP) injection with mild response, pain during daily activity, and failure to practice recreational sports. Pre-operative clinical Knee Society Score (KSS) [25] was 49, while functional KSS was 35. The patient walked for less than 5 blocks and she was unable to climb down the stairs. Pre-operatively visual analog score (VAS) was 6.

The clinical exam showed a valgus knee. No flexion contractures were observed, she actively flexed the knee for about 85 degrees. Tricompartmental pain was present at the clinical exam. The knee had a lateral wear laxity at 30 degrees of flexion, and it was stable in the antero-posterior plane.

X-rays were obtained. The X-rays showed a valgus knee (4.8 degrees of mechanical axis and 11 degrees of anatomical one) (Figure 1). The antero-posterior X-rays showed lateral osteoarthritis. Based on Kellgren–Lawrence, there was evidence of grade IV on the lateral compartment and a grade II–III on the patella-femoral joint and on the femoro-tibial compartment (Figure 2).

Spinal anesthesia and a saphenous block were performed. Pre-operative prophylactic antibiotics with intravenous 2 g of cefazoline were administered.

A 10 cm incision was made and a midvastus approach was performed to expose the joint. No tourniquet was applied at the lower limb in order to have better control of the hemostasis during the procedure, to avoid retraction of the extensor mechanism, and to better check patellofemoral tracking during patellofemoral joint arthroplasty.

The patient was scheduled for a combined UKA (lateral plus patellofemoral), but during surgery, a severe chondropathy of the medial condyle was discovered. So, due to an intact and efficient ACL in an active sporty woman, a tricompartmental UKA was performed.

All the osteophytes were removed to achieve a better balancing of the knee in flexion and extension.

For the tibio-femoral components, both medial and lateral Journey II UNI (Smith & Nephew, Watford, UK) were chosen. Journey II UNI has a J-curved femoral component to mimic the femoral condyle’s anatomic shape. Journey II patellofemoral joint (Smith & Nephew, Watford, UK), which has an asymmetric trochlear geometry, was used. All femoral components are made of Oxinium with a tibial base plate in titanium. The implant choice was due to a nickel allergy.

An 8 mm medial tibial cut was performed at 90 degrees on the coronal plane with 3 degrees of posterior slope. The lateral tibial cut was performed at 90 degrees on a coronal plane with 0 degrees of posterior slope, correcting the valgus.

After checking the two cuts in flexion and extension with spacers, the 8 mm medial and lateral femoral cutting guides were placed to perform the distal femoral cuts. With the medial and lateral spacers, the extension gap and limb alignment were checked again. Congruency of the joint line was then ensured.

The medial 3-in-1 cutting guide was positioned 2 mm under the femoral cartilage on the lateral edge of the medial femoral condyle near the notch and the cut was made. The lateral 3-in-1 femoral cutting guide was placed in a position as lateral as possible and perpendicular to the tibial cut.

The tibial plateau sizes were measured with specific instrumentation and the correct sizes were chosen, checking antero-posterior and medio-lateral distances. Tibial pegs and keels were made. Medial and lateral femoral trial components with trial bearings were positioned. Limb alignment, stability in flexion and extension, range of motion and joint line restoration were checked.

After placing the bi-uni, the patellofemoral joint was prepared.

The anterior femoral cut was performed by using the cutting guide and sizing of the trochlear trial was established finding a good compromise between the lateral femoral component and the trochlea.

Once the trochlear groove was drilled, the trochlear component was placed as large as possible in order to obtain better patellofemoral tracking, depending on the patient’s femoral anatomy and lateral femoral trail component.

The patellar cut was performed with specific instruments. The correct patellar trail size was chosen and placed as medial as possible to gain better tracking.

All the trail components were removed, and the sclerotic bone was drilled to have a better interdigitation of the cement (Figure 3).

At first, the lateral tibial plateau was cemented, then the medial one, and then their bearings were added. At the end, the femoral components were cemented. The trochlear component and the patellar one were the last to be cemented (Figure 4).

Accurate hemostasis and local infiltration anesthesia (LIA) were performed around the anterior capsule, in the hoffa pad, beneath the quadriceps tendon and in the postero-medial capsule.

Closure of the arthrotomy was performed by a barbed suture, checking at the end the patella-femoral tracking. Subcutaneous tissue was closed with an absorbable braided suture. The skin was closed with an intradermic suture. No drain was placed. Post-operative X-rays were obtained (Figure 5).

The patient was allowed to start moving the knee in flexion and extension once the anesthesia was solved. The day after surgery, she was allowed to practice weight-bearing with canes and started post-operative rehabilitation in the inpatient center. No post-operative complications were described. Two days after surgery, C-reactive protein (CRP) was 5.3 mg/dL.

The patient started walking with two canes on the first day after surgery and climbing stairs with canes in 5 days. She was dismissed from the hospital 15 days after surgery; 20 days post-operatively, the patient started walking with one cane. Five weeks after surgery, the CRP level tested was 0.88 mg/dL.

One year after surgery, the patient reported an improvement in pain and range of motion, as well as improvement in patient-reported outcomes. She gained a level 4 tegner activity scale [26] (recreational sport bicycling). At latest follow-up, the Knee Society Score (KSS) was 95, while the KSS functional score was 100. Post-operatively, the visual analogue score (VAS) was 0. X-rays at the last follow-up showed no sign of loosening or osteolysis (Figure 6).

## 3. Discussion

Traditionally, when all three compartments are involved, end-stage knee OA is treated with a total knee arthroplasty (TKA) [18]. On the other hand, in the case of isolated osteoarthritis (OA), UKA is the treatment of choice [19] using medial or lateral unicondylar or patellofemoral implants. In comparison to TKA, UKA has a quicker recovery time [27], less morbidity and more tissue sparing [19], and less blood loss [28,29].

Lombardi et al. [27] compared 103 patients treated with a UKA device to a selected group of 103 patients who underwent a cruciate retaining total knee arthroplasty (CR TKA). The CR TKA group showed a worse range of motion at discharge and longer hospital stay, as lower functional scores. No differences were found in the average time to return to work and sport, suggesting that the UKA group was allowed to a faster return to a more functional level. Return to work (RTW) and ability to sustain work by job after knee replacement were analyzed in the Clinical Outcomes in Arthroplasty, showing that most people receiving knee arthroplasty return to work, but with more difficulties in case of TKA than UKA. Among associate professional or technical occupations, RTW rates were higher in the UKA, with a knee-related job loss of 8.5% in the UKA group versus 16.7% in the TKA group, if the job involved carrying, lifting or climbing [30]. In addition in a recent review, the Early Osteoarthritis group of ESSKA (European Knee Associates section) reported a mean implant survival of up to 96.5% at a 10-year follow-up, while all the outcomes reported were improved following UKA [31].

Tricompartmental disease can be observed at the time of surgery in 17% of patients and most patients with a Kellegren–Lawrence grade >3 have an intact ACL [20]. ACL is crucial to reproduce normal joint kinematics, such as normal gait, femoral roll back and screw-home movement [32]. TKA sacrifices the ACL with a bit lower function and increased laxity [19].

Patients with concurrent medial and lateral osteoarthritis and a functional ACL may receive a primary bi-unicondylar arthroplasty (Bi-UKA) [21]. Combined partial knee arthroplasty (CPKA) is an established practice [21,22]: two UKA positioning in the same knee offer a better knee function preserving bone and cruciate. Two UKA positioning in the same knee is a highly demanding technique, offering a better knee function due to the maintenance of the essential features of native knee kinematics (femoral rollback and the screw-home mechanism) preserving bone and cruciate (Table 1).

Garner et al. [33] compared CR implants with Bi-UKA, finding a more normal gait and improved patient-reported outcomes in the Bi-UKA group.

Romagnoli et al. reported a 94.2% survival rate of Bi-UKA replacement of the two tibiofemoral compartments at 9.4 years of follow-up with improvement in knee joint range of motion and in clinical and functional KSS [21]. In a previous study, they also reported 95.2% survivorship at a 5-year follow-up of 105 gender-specific patellofemoral arthroplasties (PFA) either isolated or combined with UKA, demonstrating an improvement in knee joint range of motion and functional scores [34].

The literature reports OA progression as one of the most common reasons for revision of UKA: revisions with the addition of another UKA are described in the literature [21,22,23].

Pandit et al. [35] reported 27 knees of two-staged Bi-UKA for lateral progression of arthritis following medial UKA, with a significant improvement in functional scores and no further surgeries or revisions at the final follow-up.

Furthermore, a conversion to a tricompartmental joint replacement with retention of both cruciates was described by Rolston [19]. A bi-compartmental implant (Deuce; Smith & Nephew, Memphis, TN, USA) was added to a lateral UKA, in order to solve the disease progression in the medial and patellofemoral compartments while correcting varus deformity. They found a well-fixed lateral UKA with no sign of wear. A revision of a well-fixed UKA device can lead to significant bone loss with the need for a tibial augment and stem to reconstruct the joint and to gain base-plate stability, compromising knee function [36]. This monolithic femoral component however performed very poorly with high rates of revision for malalignment, sizing difficulties and tibial component fractures [37].

We reported to our knowledge the first case of hypoallergenic tricompartmental one-stage UKA. The surgery presented by Rolston et al. [19] showed a two-stage procedure. Their work indeed describes a revision case. Our report differs from the previous study because it is a one-stage procedure. One-stage tricompartmental UKA is a more demanding technique because it is started from a tricompartmental OA, so a worse initial joint condition. At the same time, it is difficult to make all small replacements work together not beginning from a well-functioning UKA.

Compared to TKA, UKA seems to have lesser costs [38]. In this particular case, the costs of the materials had shown to be more than TKA but, due to the strict indication, not many cases can be performed. It can be revised with primary implants due to its major bone preservation, reducing costs for future revisions.

At a one-year follow-up, the patient is satisfied. She has an improvement in knee pain and function. The KSS clinical score improved from 49 pre-operatively to 95 at the last follow-up, while the KSS functional score ranged from 35 to 100. One year postoperative X-rays showed no sign of loosening or osteolysis.

## 4. Future Directions, Clinical Implication and Lessons Learned

More studies are needed to address the real clinical impact of a tricompartmental one-stage UKA and also a comparison with TKA is needed. A tricompartmental one-tage UKA procedure allowed us to have major bone preservation for future revisions and an improvement in knee function. It is a highly demanding technique with a long learning curve.

## 5. Conclusions

A tricompartmental one-stage UKA procedure allowed us to have major bone preservation for future revisions, with primary implants and an improvement in knee joint function and proprioception due to ACL preservation with reduction of pain.

## Figures and Tables

**Figure 1 healthcare-11-02999-f001:**
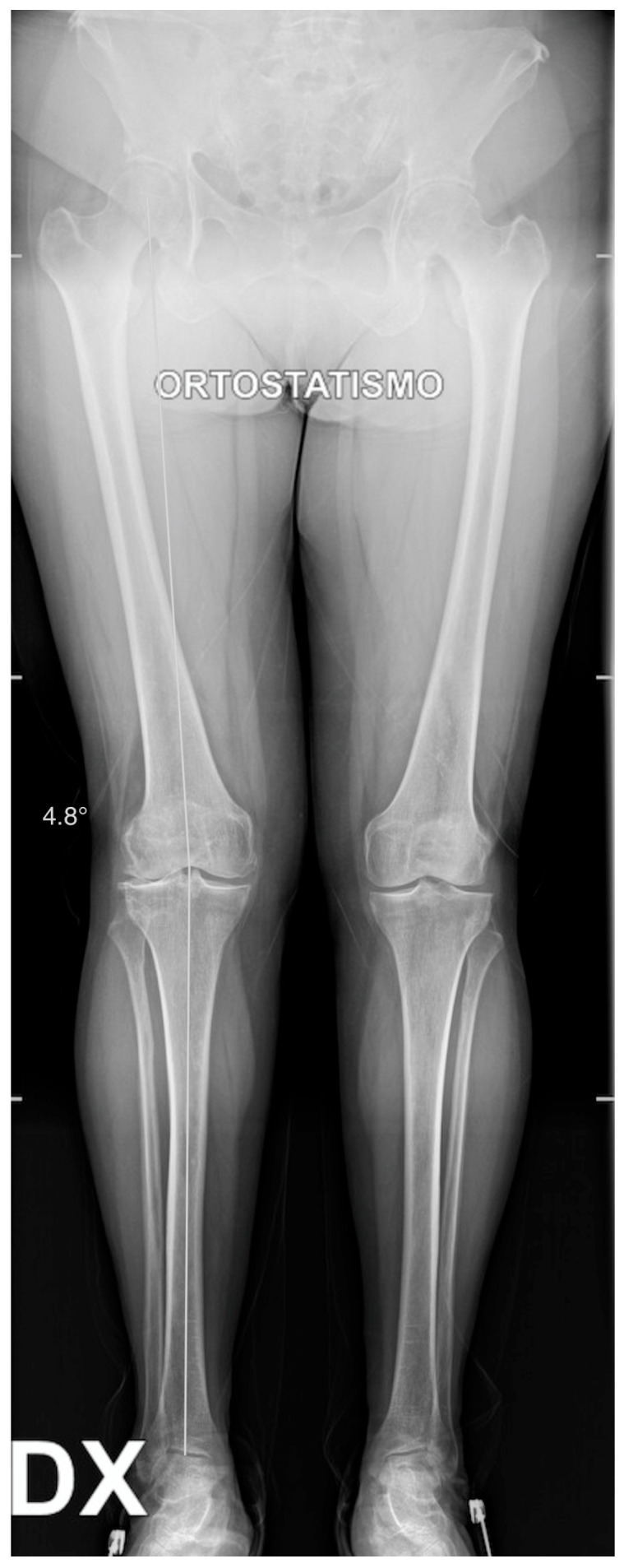
Long standing X-ray. Valgus knee: 4.8 degrees of valgus.

**Figure 2 healthcare-11-02999-f002:**
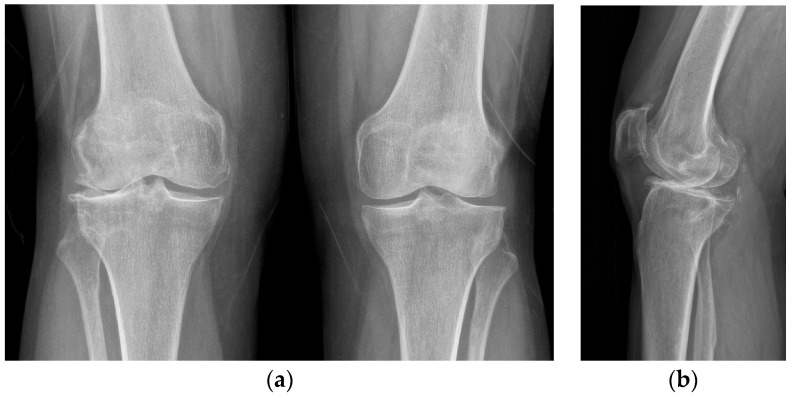
Pre-operative X-rays. (**a**) AP View: Kellgren–Lawrence grade IV on lateral compartment, grade II on medial compartment; (**b**) Lateral View: Kellgren–Lawrence grade III on patellofemoral joint.

**Figure 3 healthcare-11-02999-f003:**
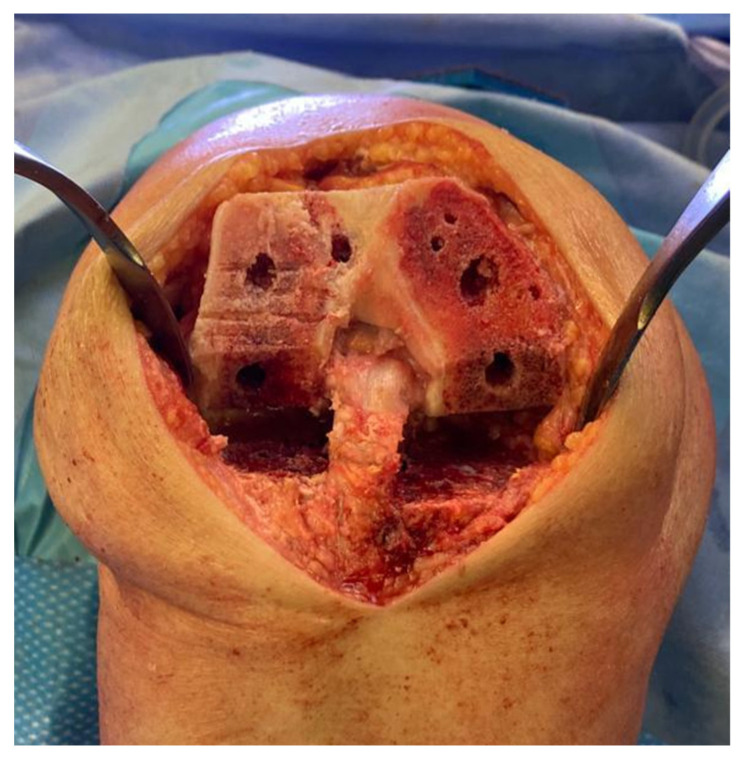
Sclerotic bone was drilled to have a better interdigitation of the cement.

**Figure 4 healthcare-11-02999-f004:**
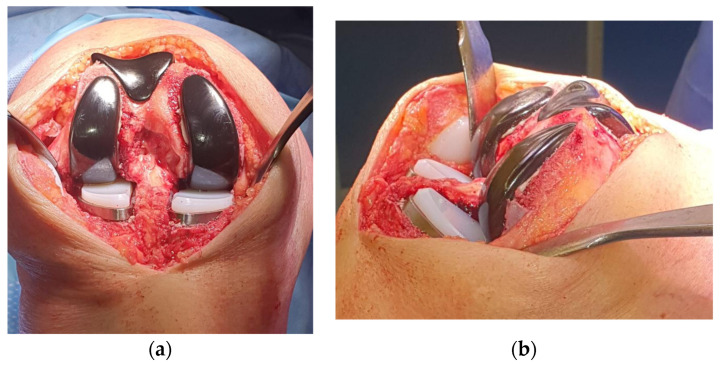
Component positioning. (**a**) Anterior view with definitive components. The two fixed bearings are on the same level to restore the correct joint line. (**b**) Lateral view with definitive components. Two different degrees of posterior slope for the tibial cuts.

**Figure 5 healthcare-11-02999-f005:**
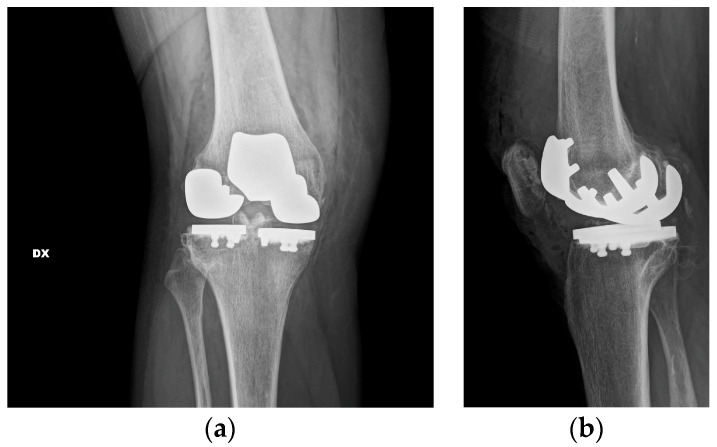
Postoperative X-rays. (**a**) AP view. (**b**) Lateral view.

**Figure 6 healthcare-11-02999-f006:**
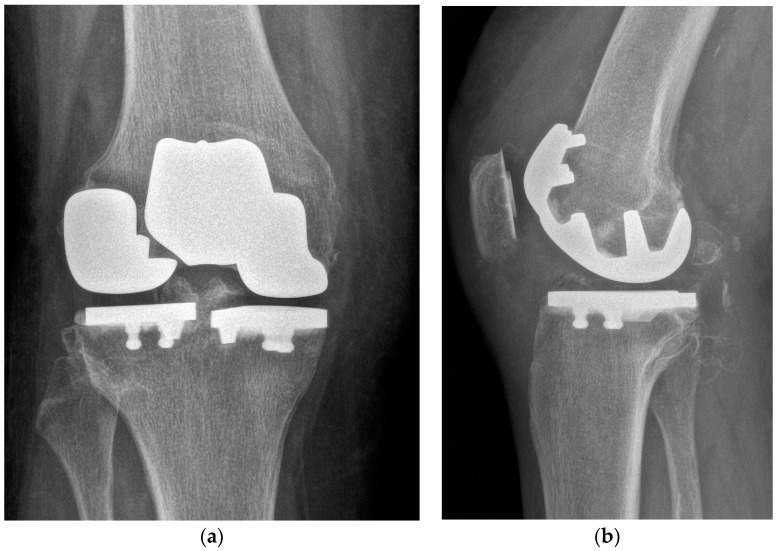
X-rays at last follow-up. (**a**) AP view. (**b**) Lateral view.

**Table 1 healthcare-11-02999-t001:** CPKA-related articles.

Title	Author	Contribution
Knee resurfacing with double unicompartmental arthroplasty: rationale, biomechanics, indications, surgical technique and outcomes [21].	Romagnoli, S., Petrillo, S. and Marullo Matteo	Recreating normal knee kinematics and function. Excellent clinical results, with a gait pattern and knee function closer to the native one compared to TKA; 94.2% of overall survival rate at 9.4 years of follow-up
Mid- to long-term follow-up of combined small implants [22].	Rossi, S.M.P., Perticarini, L., Clocchiatti, S., Ghiara, M., Benazzo, F.	Excellent clinical and radiological outcomes 91.5% survival rate at mid- to long-term follow-up
Bi-unicondylar arthroplasty: A biomechanics and clinical outcomes study [33].	Garner, A. J., Dandridge, O. W., Amis, A. A., Cobb, J. P., van Arkel, R. J.	Bi-UKA restores a more normal gait than TKA. Patients are highly satisfied and report excellent quality of life following Bi-UKA. Bi-UKA subjects reported higher OKS and EQ-5D.
Mid-merm Clinical, Functional, and Radiographic Outcomes of 105 Gender-specific Patellofemoral Arthroplasties, With or Without the Association of Medial Unicompartmental Knee Arthroplasty [34].	Romagnoli S., Marullo M.	Improvement in knee joint range of motion, clinical and functional Knee Society Score at mid-term follow-up; 95.2% survival rate at 5.5 years follow-up.

## Data Availability

Data are contained within the article.

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
