# Peer review of "One-Stage Tricompartmental Hypoallergenic UKA for Tricompartmental Osteoarthritis: A Case Report"

_healthcare, 2023, doi:10.3390/healthcare11222999_

Round 1

Reviewer 1 Report

Comments and Suggestions for Authors

The authors have produced an interesting and useful case report describing a one-stage hypoallergenic tricompartmental knee arthroplasty with improved clinical score and no signs of early failure at last follow-up.

However, I would like to make a few observations before recommending their work for publication.

1. Please, could you clarify the type of study in the title?

2. The Introduction section needs to be developed a little more, commenting briefly on the conservative treatment. I recommend that the authors to comment on the conservative treatment that can be very useful in mild to moderate OA. I recommend using and mentioning the following quality papers of studies investigating the use of Dry Needling and a 3-month program of Therapeutic Exercise: doi:10.1093/pm/pnz036.

3. Please verify your research follows and cite the CARE criteria for Case Reports.

4. Could the authors add a Graphical Abstract?

5. There is an interesting work that develops to identify the differences in blood investigations between total hip and total knee replacement. I recommend the authors to discuss it: 10.1097/TGR.0000000000000337

6. In the Discussion section, could you add a section on "Future Directions, Clinical Implications and Lessons Learned"?

7. The authors should revise the wording of their conclusions, since their study is not comparative.

Comments on the Quality of English Language

No comments

Author Response

Dear Reviewer,

Response to Reviewer 1 Comments

1. Summary

Thank you very much for taking the time to review this manuscript.

The suggestions offered by the reviewers have been immensely helpful, and we also appreciate your insightful comments on revising several aspects of the paper. We believe have resulted in an improved revised manuscript, which you will find uploaded.

2. Questions for General Evaluation

Reviewer’s Evaluation

Response and Revisions

Does the introduction provide sufficient background and include all relevant references?

Must be improved

We improved introduction

Are all the cited references relevant to the research?

Can be improved

We added citations

Is the research design appropriate?

Yes

Thank you

Are the methods adequately described?

Can be improved             

We improved some parts of the paper

Are the results clearly presented?

Can be improved

We added new pieces of information on the case

Are the conclusions supported by the results?

Yes

Thank you

 3. Point-by-point response to Comments and Suggestions for Authors

Comments 1: Please, could you clarify the type of study in the title?

Response 1: Thank you for pointing this out. We agree with this comment. Therefore, we have changed the title: One-Stage-Tricompartmental Hypoallergenic UKA for Tricompartmental Osteoarthritis: a case report

Comments 2: The Introduction section needs to be developed a little more, commenting briefly on the conservative treatment. I recommend that the authors to comment on the conservative treatment that can be very useful in mild to moderate OA. I recommend using and mentioning the following quality papers of studies investigating the use of Dry Needling and a 3-month program of Therapeutic Exercise: doi:10.1093/pm/pnz036.

Response 2: Agree. Thanks for your kind reminders. We added a little paragraph in the introduction (Lines 47-53): Conservative treatment can be very useful in mild to moderate knee osteoarthritis. Pain and alterated physical function can be treated effectively with physiotherapy through therapeutic exercise, based on quadriceps strengthening and aerobic exercise, improving range of motion and proprioception. Dry needling (DN) used in combination with physiotherapy can produce significant improvement in pain, range of motion, and function in patients with persistent pain after TKA. To improve pain and function in painful knee OA electrical DN in a manual therapy can be added to the exercise program.[11]

Comment 3. Please verify your research follows and cite the CARE criteria for Case Reports.

Response 3: Thank you for your suggestion. We followed Care criteria and we added a line on the paper (line 89). “CARE (CAse REport) criteria for case reports were followed. [25]”

Comment 4. Could the authors add a Graphical Abstract?

Response 4 Thank you so much for your suggestion and pointing this out. It is a very interesting point. It is very hard to resume the work essence in a graphical abstract.

Comment 5. There is an interesting work that develops to identify the differences in blood investigations between total hip and total knee replacement. I recommend the authors to discuss it: 10.1097/TGR.0000000000000337

Response 5 Thank you for the kind suggestion, we discussed it in line 214  “less blood loss [29,30].”

Comment 6. In the Discussion section, could you add a section on "Future Directions, Clinical Implications and Lessons Learned"?

Response 6 Thanks for your kind reminders. We added a paragraph at the end of the discussion. Lines 271-275 “4. Future Directions, Clinical Implication and Lessons Learned

More studies are needed to address the real clinical impact of a tricompartmental one stage UKA and also a comparison with TKA is needed. A tricompartmental one stage UKA procedure allowed us to have a major bone preservation for future revisions and an improvement in knee function. It is a highly demanding technique with a long learning curve.”

Comment 7. The authors should revise the wording of their conclusions since their study is not comparative.

Response 7 Thanks for your kind suggestion. We modified the conclusion (Lines 277-279)

“A tricompartmental one stage UKA procedure allowed us to have a major bone preservation for future revisions with primary implants and an improvement in knee joint function and proprioception due to ACL preservation with reduction of pain. “

4. Response to Comments on the Quality of English Language

Point 1: No comments

Response 1: Thank you.

Reviewer 2 Report

Comments and Suggestions for Authors

Osteoarthritis has the potential to impact various compartments of the knee. While total knee arthroplasty (TKA) serves as a solution for advanced OA, unicondylar knee arthroplasty (UKA) emerges as a primary treatment for selective cases, leading to a swifter recovery with reduced complications. The research presented by Andrea and colleagues introduces a unique one-stage hypoallergenic tricompartmental UKA, showcasing enhanced clinical results without any indications of premature failure. The study undoubtedly piques interest, but several aspects require further clarification before it can be deemed suitable for publication:

  1. 1. The term "hypoallergenic" prominently features in the title. Could the authors elucidate its significance and relevance to this study? It would be beneficial to delve deeper into this in the introduction and discussion sections.
  2. 2. Presenting pre-surgery MRI results would enhance the paper's depth. Such results would effectively display the extent of cartilage and ligament damage in the patients.
  3. 3. While it's established that unicompartment UKA offers a speedier recovery than TKA, it would be insightful if the authors could draw a comparison between the tricompartmental UKA implemented in this study and the traditional unicompartment UKA.
  4. 4. Including specific post-operative data, such as the ESR, CRP levels, and alterations in pain scores pre and post-surgery, would enrich the study's findings and provide a more comprehensive view of the patient's recovery trajectory.
Comments on the Quality of English Language

The passage is generally well-written and contains specialized medical terminology appropriate for a scientific or clinical audience.

Author Response

Dear Reviewer,

Response to Reviewer 2 Comments

1. Summary

Thank you very much for taking the time to review this manuscript.

The suggestions offered by the reviewers have been immensely helpful, and we also appreciate your insightful comments on revising several aspects of the paper. We believe have resulted in an improved revised manuscript, which you will find uploaded.

2. Questions for General Evaluation

Reviewer’s Evaluation

Response and Revisions

Does the introduction provide sufficient background and include all relevant references?

Must be improved

We improved introduction

Are all the cited references relevant to the research?

Must be improved

We added references

Is the research design appropriate?

Can be improved

We improved the paper with new information on the case

Are the methods adequately described?

Can be improved             

We improved some parts of the paper

Are the results clearly presented?

Can be improved

We added new pieces of information on the case

Are the conclusions supported by the results?

Must be improved

We changed the conclusion adding a new paragrahph

 3. Point-by-point response to Comments and Suggestions for Authors

Comment 1: The term "hypoallergenic" prominently features in the title. Could the authors elucidate its significance and relevance to this study? It would be beneficial to delve deeper into this in the introduction and discussion sections.

Response 1: Thank you for pointing this out. The hypoallergenic term features the title in order to specify the type of UKA used. Some UKA implant designs differ from hypoallergic to allergic one. This type of UKA does no differ between the two types, gaining the same level of function.

Comment 2: Presenting pre-surgery MRI results would enhance the paper's depth. Such results would effectively display the extent of cartilage and ligament damage in the patients.

Response 2: Thank you for pointing this out. We do not request MRI in the case of UKA on a daily basis. We usually made our patient selection on the clinical exam and x-rays.

Comment 3. While it's established that unicompartment UKA offers a speedier recovery than TKA, it would be insightful if the authors could draw a comparison between the tricompartmental UKA implemented in this study and the traditional unicompartment UKA.

Response 3: This is a very interesting point, thank you for the suggestion. This type of surgery is a bridge between TKA and UKA. UKA is less invasive and has a better postoperative function due to a less invasive approach and major preservation of the naive knee. More cases are needed to have a better comparison between UKA and the tricompartmental UKA.

Comment 4.  Including specific post-operative data, such as the ESR, CRP levels, and alterations in pain scores pre and post-surgery, would enrich the study's findings and provide a more comprehensive view of the patient's recovery trajectory.

Response 4 Thank for you your suggestion. We do not routinely dose ESR. As suggested, we added more information to patient score, adding VAS score and post-operative CRP value. Lines 98-99 “Pre-operatively visual analogue score (VAS) was 6.”, “Two days after surgery C-reactive protein (CRP) was 5.3 mg/dl.” 192-193 , “Five weeks after surgery CRP level tested was 0.88 mg/dl.” 196-197, “Post-operatively visual analogue score (VAS) was 0. “201-202.

4. Response to Comments on the Quality of English Language

Point 1: The passage is generally well-written and contains specialized medical terminology appropriate for a scientific or clinical audience.

Response 1: Thank you.

Reviewer 3 Report

Comments and Suggestions for Authors

I would like to express certain concerns regarding the present report:

1.     The paragraphs, I find, are rather succinct throughout the manuscript. It would be advantageous to amalgamate some of them for a more cohesive narrative.

2.     It is advisable to eschew gratuitous capitalisation of letters throughout the manuscript.

3.     It is suggested to expound further on the merits and drawbacks of CPKA, as per line 73.

4.     On line 94, kindly furnish the preoperative anatomic and mechanical angles.

5.     I would like to request the provision of postoperative long-standing X-rays.

6.     A tabular summary encompassing all CPKA-related articles would be most beneficial.

7.     A detailed comparative analysis between the present study and reference 23 is sought. Kindly elucidate, as tricompartmental UKA has previously been documented by Rolston et al. What, in your esteemed opinion, distinguishes the current study and renders it worthy of publication?"

Comments on the Quality of English Language

The standard of English presented is modest, leaving ample space for refinement and enhancement.

Author Response

Dear Reviewer,

Response to Reviewer 3 Comments

1. Summary

Thank you very much for taking the time to review this manuscript.

The suggestions offered by the reviewers have been immensely helpful, and we also appreciate your insightful comments on revising several aspects of the paper. We believe have resulted in an improved revised manuscript, which you will find uploaded.

2. Questions for General Evaluation

Reviewer’s Evaluation

Response and Revisions

Does the introduction provide sufficient background and include all relevant references?

Can be improved

We improved introduction

Are all the cited references relevant to the research?

Can be improved

We added references

Is the research design appropriate?

Can be improved

We improved the paper with new information on the case

Are the methods adequately described?

Can be improved             

We improved some parts of the paper

Are the results clearly presented?

Can be improved

We added new pieces of information on the case

Are the conclusions supported by the results?

Can be improved

We changed the conclusion

 3. Point-by-point response to Comments and Suggestions for Authors

 Comment 1: The paragraphs, I find, are rather succinct throughout the manuscript. It would be advantageous to amalgamate some of them for a more cohesive narrative.

Response 1: Thank you for the suggestion. We amalgamate some of the paragraphs to have a more cohesive narrative.

Comment 2: It is advisable to eschew gratuitous capitalisation of letters throughout the manuscript.

Response 2: Thank you for suggestion. We revised it.

Comment 3.   It is suggested to expound further on the merits and drawbacks of CPKA, as per line 73. 

Response 3: Thank you for your suggestion. We explained the concept in lines 76-79. “two UKA positioning in the same knee is a highly demanding technique, offering a better knee function due to the maintenance of the essential features of native knee kinematics (femoral rollback and the screw-home mechanism) preserving bone and cruciate”

Comment 4. On line 94, kindly furnish the preoperative anatomic and mechanical angles.

Response 4 Thank you for your kind suggestion. We added the angles. Lines 104-105

“The X-rays showed a valgus knee (4.8 degrees of mechanical axis and 11 degrees of anatomical one) (Figure 1).”

Comment 5. I would like to request the provision of postoperative long-standing X-rays.

Response 5. Thank you so much for your suggestion. It is a very interesting point. In our clinic, we do not routinely take postoperative standing x-rays. On the other hand, all the postoperative X-rays are obtained in standing position.

Comment 6. A tabular summary encompassing all CPKA-related articles would be most beneficial.

Response 6 Thanks for your kind reminders. We added a tabular summary of CPKA related article (Table 1). Lines 80-82

Title

Author

Contribution

Knee resurfacing with double unicompartimental arthroplasty: rationale, biomechanics, indications, surgical technique and outcomes

Romagnoli S, Petrillo S. and Marullo Matteo

Recreating normal knee kinematics and function

Excellent clinical results, with a gait pattern and knee function closer to the native one compared to TKA

94.2% of overall survival rate at 9.4 years of follow-up

Mid- to long-term follow-up of combined small implants

 Rossi S.M.P, Perticarini L, Clocchiatti s, Ghiara M, Benazzo F.

Excellent clinical and radiological outcomes 

91.5 % survival rate at mid- to long-term follow-up

Bi-unicondylar arthroplasty

A biomechanics and clinical outcomes study

Garner A. J., Dandridge
O. W.,Amis A. A., Cobb
J. P. , van Arkel
R. J.

Bi-UKA restores a more normal gait than TKA.

Patients are highly satisfied and report excellent quality of life following Bi-UKA.

Bi-UKA subjects reported higher OKS and EQ-5D .

Mid-Term Clinical, Functional, and Radiographic Outcomes of 105 Gender-Specific Patellofemoral Arthroplasties, With or Without the Association of Medial Unicompartmental Knee Arthroplasty

Romagnoli S., Marullo M.

Improvement in knee joint range of motion, clinical and functional Knee Society Score at mid-term follow-up

95.2% survival rate at 5.5 years follow up

Comment 7.   A detailed comparative analysis between the present study and reference 23 is sought. Kindly elucidate, as tricompartmental UKA has previously been documented by Rolston et al. What, in your esteemed opinion, distinguishes the current study and renders it worthy of publication?"

Response 7. Thank you for your kind suggestion. To our knowledge, Rolston at all is the only article published on a tricompartmental UKA, but the surgery presented in the article showed a two-stage procedure. Their work indeed describes a revision case. Our report differs from the previous study because it is a one stage procedure. One stage tricompartmental UKA is a more demanding technique because it is started from a tricompartmental OA, so a worse initial joint condition. At the same time, it has the difficulty to make all small replacements work together not beginning from a well-functioning UKA.

4. Response to Comments on the Quality of English Language

Point 1: The standard of English presented is modest, leaving ample space for refinement and enhancement.

Response 1: Thank you for pointing this out. We made some corrections.

Round 2

Reviewer 1 Report

Comments and Suggestions for Authors

The current version of the manuscript has been improved, and currently I recommend its publication.

Comments on the Quality of English Language

No comments

Author Response

Dear Reviewer,

Response to Reviewer 1 Comments

1. Summary

Thank you very much for taking the time to review this manuscript. We appreciate the careful review and constructive suggestions.

2. Questions for General Evaluation

Reviewer’s Evaluation

Response and Revisions

Does the introduction provide sufficient background and include all relevant references?

Yes

Thank you

Are all the cited references relevant to the research?

Yes

Thank you

Is the research design appropriate?

Yes

Thank you

Are the methods adequately described?

Yes

Thank you

Are the results clearly presented?

Yes

Thank you

Are the conclusions supported by the results?

Yes

Thank you

 3. Point-by-point response to Comments and Suggestions for Authors

Comments 1: The current version of the manuscript has been improved, and currently I recommend its publication.

Response 1: Thank you very much

 4. Response to Comments on the Quality of English Language

Point 1: No comments

Response 1: Thank you very much

Reviewer 2 Report

Comments and Suggestions for Authors

Having carefully reviewed the revised version of the manuscript, I am pleased to note significant improvements over the initial submission. The authors have evidently made a commendable effort in addressing the concerns raised during the previous review, resulting in a more robust and coherent presentation of their findings. I believe that the manuscript is now in a form that is suitable for publication.

Author Response

Dear Reviewer,

Response to Reviewer 2 Comments

1. Summary

Thank you very much for taking the time to review this manuscript. We appreciate the careful review and constructive suggestions.

2. Questions for General Evaluation

Reviewer’s Evaluation

Response and Revisions

Does the introduction provide sufficient background and include all relevant references?

Yes

Thank you

Are all the cited references relevant to the research?

Yes

Thank you

Is the research design appropriate?

Yes

Thank you

Are the methods adequately described?

Yes

Thank you

Are the results clearly presented?

Yes

Thank you

Are the conclusions supported by the results?

Yes

Thank you

 3. Point-by-point response to Comments and Suggestions for Authors

Comments 1: Having carefully reviewed the revised version of the manuscript, I am pleased to note significant improvements over the initial submission. The authors have evidently made a commendable effort in addressing the concerns raised during the previous review, resulting in a more robust and coherent presentation of their findings. I believe that the manuscript is now in a form that is suitable for publication.

Response 1: Thank you very much

Reviewer 3 Report

Comments and Suggestions for Authors

The revision has greatly improved the manuscript. I offer some residual suggestions below:

1.     Response 7 is highly commendable. I recommend integrating the paragraph into the Discussion section.

2.     I propose relocating Table 1 and lines 84-86 to the Discussion section.

3.     Lines 87-88 in the Introduction section should be merged together. Similarly, endeavour to amalgamate brief paragraphs within the Discussion section for improved coherence.

Comments on the Quality of English Language

The English quality has significantly improved following the revision process.

Author Response

Dear Reviewer,

Response to Reviewer 3 Comments

1. Summary

Thank you very much for taking the time to review this manuscript. We appreciate the careful review and constructive suggestions. Please find the detailed responses below and the corresponding corrections.

2. Questions for General Evaluation

Reviewer’s Evaluation

Response and Revisions

Does the introduction provide sufficient background and include all relevant references?

Yes

Thank you.

Are all the cited references relevant to the research?

Yes

Thank you.

Is the research design appropriate?

Yes

Thank you.

Are the methods adequately described?

Yes

Thank you.

Are the results clearly presented?

Yes

Thank you.

Are the conclusions supported by the results?

Yes

Thank you.

 3. Point-by-point response to Comments and Suggestions for Authors

 Comments 1: 1.   Response 7 is highly commendable. I recommend integrating the paragraph into the Discussion section.

Response 1: Thank you. We integrated the paragraph into the discussion section.

Lines 258-263 “The surgery presented by Rolston et al[24] showed a two-stage procedure. Their work indeed describes a revision case. Our report differs from the previous study because it is a one-stage procedure. One-stage tricompartmental UKA is a more demanding technique because it is started from a tricompartmental OA, so a worse initial joint condition. At the same time, it has the difficulty to make all small replacements work together not beginning from a well-functioning UKA.”

Comments 2.    I propose relocating Table 1 and lines 84-86 to the Discussion section.

Response 2: Thank you for your kind suggestion. We have, accordingly, modified lines in the introduction to emphasize this point. We added lines and table 1 to the discussion. Lines 77, 226-233.

Comments 3.     Lines 87-88 in the Introduction section should be merged together. Similarly, endeavour to amalgamate brief paragraphs within the Discussion section for improved coherence.

Response 3: Thank you for your kind suggestion. We have, accordingly, revised the introduction. Lines 82-83 “In our case report we describe a one-stage hypoallergenic tricompartmental UKA following CARE (CAse REport) criteria [25].”

We also merged some paragraphs to obtain more consistency. Lines 200-202, 219-222,248.

 4. Response to Comments on the Quality of English Language

Point 1: The English quality has significantly improved following the revision process.

Response 1: Thank you.